# A Data Augmentation Method for Skeleton-Based Action Recognition with Relative Features

**Junjie Chen** [1] , **Wei Yang** [2] , **Chenqi Liu** [3,*] **and Leiyue Yao** [1]

1   School of Information Engineering, Nanchang University, Nanchang 330031, China; junjie_chen2020@163.com (J.C.); leiyue_yao@163.com (L.Y.)
2   The Center of Collaboration and Innovation, Jiangxi University of Technology, Nanchang 330031, China; yang.wei@163.com
3   Network Information Center, Nanchang University, Nanchang 330031, China
*   Correspondence: chenqi_liu@163.com

**Abstract:** In recent years, skeleton-based human action recognition (HAR) approaches using convolutional neural network (CNN) models have made tremendous progress in computer vision applications. However, using relative features to depict human actions, in addition to preventing overfitting when the CNN model is trained on a few samples, is still a challenge. In this paper, a new motion image is introduced to transform spatial-temporal motion information into image-based representations. For each skeleton sequence, three relative features are extracted to describe human actions. The three relative features are consisted of relative coordinates, immediate displacement, and immediate motion orientation. In particular, the relative coordinates introduced in our paper not only depict the spatial relations of human skeleton joints but also provide long-term temporal information. To address the problem of small sample sizes, a data augmentation strategy consisting of three simple but effective data augmentation methods is proposed to expand the training samples. Because the generated color images are small in size, a shallow CNN model is suitable to extract the deep features of the generated motion images. Two small-scale but challenging skeleton datasets were used to evaluate the method, scoring 96.59% and 97.48% on the Florence 3D Actions dataset and UTkinect-Action 3D dataset, respectively. The results show that the proposed method achieved a competitive performance compared with the state-of-the-art methods. Furthermore, the augmentation strategy proposed in this paper effectively solves the overfitting problem and can be widely adopted in skeleton-based action recognition.

**Keywords:** skeleton-based human action recognition; relative coordinate; data augmentation; motion image

## 1. Introduction

In recent years, HAR has received increasing attention in the field of computer vision. Because it has a wide range of industrial applications, such as human computer interaction, smart video surveillance, and health care [1]. However, there are endemic problems, such as action diversity, partial occlusion of subjects, which hinder the development and commercial application of HAR.

The mainstream applications for HAR use RGB images or videos with the aim of simulating human vision. Motion history images and optical flows are two commonly used motion images for color-image-based methods. In the literature [2], Bobick et al. adopted motion-energy images and motion-history images to separately store motion locations and motion intensities. To produce better video representations for HAR, a number of researchers began to use a two-stream CNN to capture appearance features and motion features, and fused the classification scores of the two streams to generate the final result [3]. In addition, a few approaches employed static images; for example, Herath et al. [4] proposed a novel still image that contained optical flows and "dynamic

images" of temporal representations. Although RGB data can provide rich scene contexts and are easy to collect, RGB data are sensitive to viewpoint, background, and illumination conditions [5].

With the development of depth cameras, depth images based on HAR have received the attention of researchers. Compared with RGB images, depth images more easily segment foreground human subjects even though the scene is cluttered and can provide more geometric information [1,5]. In [6], the authors created three dynamic depth images to fully exploit the spatial, temporal, and structural information in a depth sequence for action recognition at different timescales. To better exploit 3D characteristics, Xiao et al. [7] densely projected the raw depth video to different imaging viewpoints within a 3D space to construct multiview dynamic images. Nevertheless, depth images also have drawbacks; for example, they are limited by workable distances and are vulnerable to noise [1,5].

With the increasing use of algorithms to capture the 3D coordinates of human skeletal joints, the skeleton-based human action recognition research area is emerging. Compared with former methods, skeleton data can provide 3D structural and pose location information, making motion representation simpler and more informative. Furthermore, skeleton data have a lower computational load and are more robust when dealing with changing conditions, such as different body scales and motion speeds [1,5]. Similar to color-image-based methods, skeleton-based methods can also be divided into two categories, handcrafted feature-based methods and deep learning feature-based methods. To form a final feature descriptor, HAR using handcrafted features has two stages, feature extraction and feature representation. In the former stage, various sorts and varieties of motion features are proposed, such as the relative coordinates and angles between joints. The feature representation stage may simply concatenate the features that are obtained in the first stage or use a machine learning technique to obtain the final features. Skeleton-based HAR of handcrafted feature-based methods is shown in [8–11]. Different from handcrafted feature-based methods, deep learning feature-based methods can automatically extract and learn multiple high-level features from video sequences or skeleton data by constructing multiple connected convolutional layer stacks. In recent years, HAR with deep learning feature-based methods has been reported in articles [12–14]. However, deep CNNs require a large number of labelled training samples to obtain peak performance. So, deep learning feature-based methods cannot provide noteworthy results for small sample datasets [1].

Although skeleton-based HAR methods have been greatly improved, some deficiencies still cannot be ignored. For example, in the literature [9,10], the original coordinates were directly utilized to depict human motions, leading to unsatisfactory results. Because absolute motion data are easily affected by variations in viewpoint and human appearance. Although the authors [14] obtained excellent accuracy, the methods that use absolute motion data will inevitably meet dataset-based or camera-based problems. In [11], the authors adopted a sequence of joint angles to label human motions. However, only using geometrical information is insufficient to describe an action. For example, "sitting down" and "standing up" have similar spatial motion data but have different motion data in the temporal dimension. In addition, when training CNN models on small-scale datasets, overfitting is a fatal issue that needs special attention.

To address the mentioned issues, we propose a novel skeleton-based action recognition method using a CNN model. In our proposed method, the spatial-temporal motion data of human action are encoded into an image-based representation. According to [15], the relative features can provide a view-invariant representation of human actions. Hence, three relative features are extracted, frame-by-frame, to describe human motions more accurately. The three relative features are consisted of relative coordinates, immediate displacement, and immediate motion orientation. In the following step, these feature data are stored in feature matrices and are normalized to (0, 255) to fit the storing requirement of the RGB channels and transformed into a color image. Furthermore, to cope with varying-length skeleton sequences, an effective skeleton sequence refinement strategy is utilized to align action sequences. By this way, the generated motion images have consistent spatial

sizes. In addition, in response to the issue of the limited size of training samples, a data augmentation strategy is proposed to generate samples and prevent the CNN model from overfitting. Finally, a shallow CNN model is sufficient to efficiently extract deep features because of the small size of our proposed skeleton-based motion image. Moreover, the proposed CNN model guarantees the high effectiveness of our method.

The main research contributions of this paper are summarized as follows:

(1). A relative coordinate is introduced to describe human motions, which not only depicts the spatial relations of human skeleton joints but also provides long-term temporal characteristics.

(2). An effective skeleton sequence refinement strategy, including an average linear interpolation algorithm and a randomly discard frame approach, is proposed to address varying-length action sequences.

(3). A human action-oriented data augmentation strategy with a strong generalization capacity is proposed to generate action samples for recognition accuracy improvements and overfitting prevention.

The remaining parts of this paper are structured as follows: In Section 2, the related works in the field of action recognition using skeleton data are reviewed. Then, our proposed HAR method is elaborated in Section 3. Section 4 presents experiments on the two small datasets. Finally, the conclusion of the paper is given in Section 5.

## 2. Related Work

In recent years, skeleton-based HAR has received tremendous attention in the research community. Compared with color images and depth images, skeleton data are more robust to changing conditions and occlusions [5]. Most studies based on skeleton data adopt different feature extraction techniques to preprocess the skeleton data, and then motion features are expressed as matrices or encoded into images to serve as the input to the CNN models for training purposes.

Li et al. [16] selected three joints as reference points to obtain new joint coordinates and transformed them into RGB images for fine-tuning the pretrained VGG19 model. Huynh-The et al. [17] designed an image encoder to transform the original coordinate data to image-formed data. Nevertheless, both methods ignored the temporal information, which is important in describing human motions. To consider temporal dimension information, in [18,19], the authors encoded spatial-temporal information to motion images based on joint trajectory maps (JTMs). Although the two methods have added temporal features and achieved competitive results, it inevitably loses important information due to the issue of overlapping joint trajectories.

To represent human actions more accurately, various advanced features and image coding methods have been proposed. Zhang et al. [20] summarized eight geometric features involving joint-joint distance and joint-joint orientation. In [21], the authors introduced a novel image coding technique to transform joint distance and orientation into color pixels. Moreover, a data augmentation method was used for information enhancement. Liu et al. [22] proposed a group of kinetic skeleton features to obtain the spatial-temporal characteristics, and a 1D CNN model was introduced to identify human actions.

Although skeleton-based HAR has made great improvements in accuracy, there are still two major challenges. First, utilizing the original coordinates, or only the spatial information, to describe human motion is not an ideal approach. Second, insufficient training samples prevent deep neural networks from efficiently learning deep features. Consequently, in our method, several relative features are utilized to represent human motions and a data augmentation strategy is proposed for information enrichment and overfitting prevention.

## 3. Method

In this section, we introduce our HAR method in detail. It can be divided into three steps. First, given a skeleton sequence, 15 joints are selected to eliminate redundant

information and ensuring the accuracy of classification. Then, a data augmentation strategy is proposed to generate training samples to improve recognition accuracy and prevent the CNN model from overfitting. In particular, the action sequence refinement strategy is adopted to normalize the varying-length action samples to a fixed value. By this method, generated motion images contain the same size. In the following step, three relative features are calculated frame-by-frame to represent human motions. Specifically, the relative coordinates proposed in our method not only depict the spatial relations of human skeleton joints but also provide long-term temporal information. These spatial-temporal motion features are normalized to (0, 255) to meet the storage requirements of the RGB channels and are transformed into a color motion image. Consequently, the generated motion images are small in size and simple in structure. A shallow CNN model with five hidden layers is sufficient to extract the deep features of the generated motion for action recognition purposes. The overall flowchart of our proposed method is illustrated in Figure 1.

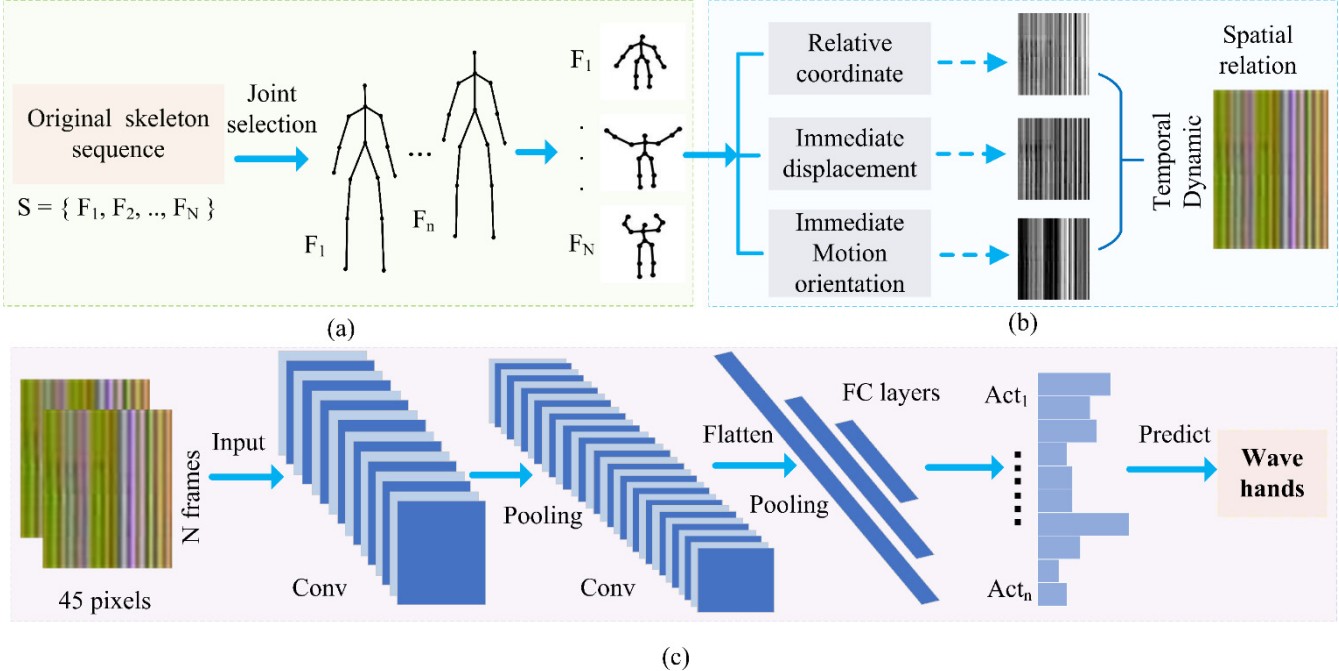

**Figure 1.** The overall flowchart of our method for action recognition. (**a**) Extract 15 joints for each original skeleton sequence, and a data augmentation strategy is proposed to enlarge training samples and prevent overfitting. (**b**) Calculate three relative features to represent human actions and then encode these features into a color image. (**c**) Input the generated motion images into a shallow CNN model for HAR.

### 3.1. Joint Selection

In the selection of human body joints, 15 joints are used in the proposed method. Although Kinect 2.0 can obtain the coordinates of 25 joints, not all joints have to be used for recognizing human actions. As mentioned in [23,24], human actions are mainly accomplished by the cooperation of the limbs and trunk, which means that some joints are informative for recognizing human actions, whereas the rest are redundant. For example, except for some subtle actions such as sign language or hand poses, the motion information of hand joints is redundant. Moreover, those redundant joints may cause noise and lead to low recognition accuracy.

Therefore, for the purpose of eliminating redundant information and ensuring the accuracy of classification, the 15 joints labelled in Figure 2 are adopted in the proposed method.

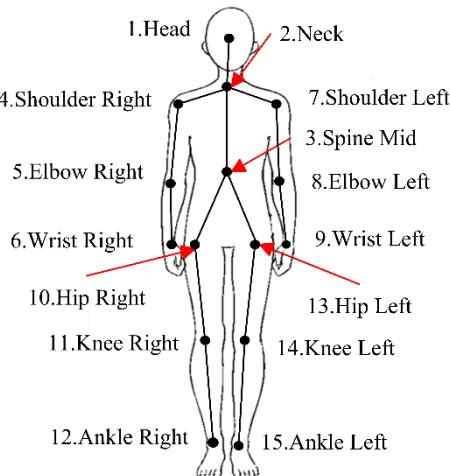

**Figure 2.** The 15 skeleton joints select in our methods.

### 3.2. Data Augmentation

In deep learning, CNN models need a considerable number of samples for training to obtain satisfactory results, or else there may be overfitting, and the model's generalization ability may drastically decrease. However, for certain actions, it is difficult to obtain a quantity of available and quality samples, so data augmentation methods are used to enlarge the dataset. Although many data augmentation methods are designed for images, such as scaling and cropping, they are not suitable for motion images. For motion images, the continuity of pixel semantics represents the consecutiveness of the motion information. A random scaling or cropping operation will destroy the authenticity of the spatial-temporal features of the actions and will inevitably result in incorrect recognition or classification.

Therefore, a data augmentation strategy is proposed to generate skeleton samples and prevent the CNN model from overfitting. The strategy consists of mimicking a person performing an action with different speeds, mimicking a certain action that was executed by people of different sizes and mimicking a certain action that was conducted in different places in the field of view. The detail of the method is described as follows.

#### 3.2.1. Mimicking a Person to Operate an Action with Different Speeds

It is known that images input into CNN models for training should remain the same size. Nevertheless, in public datasets, there are a few action sequences with consistent frame number. This means that varying-length action samples cannot be directly transformed to trainable images. Therefore, an action sequence refinement strategy is introduced to normalize the varying-length action samples to a fixed value and preserve the overall action information. The proposed strategy includes an average linear interpolation algorithm and randomly discards frame approach. In Formula (1), assume that the matrix (a) represents an original skeleton sequence with $K$ joints and $n$ frames,

$$
\begin{bmatrix}
J_{1,1} & \cdot & \cdot & J_{k,1} \\
J_{1,2} & \cdot & \cdot & J_{k,2} \\
\cdot & \cdot & \cdot & \cdot \\
\cdot & \cdot & \cdot & \cdot \\
J_{1,n} & \cdot & \cdot & J_{k,n}
\end{bmatrix}
\quad
\begin{bmatrix}
J_{1,1} & J_{2,1} & \cdot & \cdot & J_{k,1} \\
\cdot & \cdot & & \cdot & \cdot \\
J_{1,m} & J_{2,m} & \cdot & \cdot & J_{k,m} \\
N_1 & N_2 & \cdot & \cdot & N_k \\
J_{1,m+1} & J_{2,m+2} & \cdot & \cdot & J_{k,m+1} \\
\cdot & \cdot & \cdot & \cdot & \cdot \\
J_{1,n} & J_{2,n} & \cdot & \cdot & J_{k,n}
\end{bmatrix}
\quad
\begin{bmatrix}
J_{1,1} & J_{2,1} & \cdot & \cdot & J_{k,1} \\
\cdot & \cdot & & \cdot & \cdot \\
J_{1,m-1} & J_{2,m-1} & \cdot & \cdot & J_{k,m-1} \\
J_{1,m+1} & J_{2,m+1} & \cdot & \cdot & J_{k,m+1} \\
\cdot & \cdot & \cdot & \cdot & \cdot \\
\cdot & \cdot & \cdot & \cdot & \cdot \\
J_{1,n} & J_{2,n} & \cdot & \cdot & J_{k,n}
\end{bmatrix}
\tag{1}
$$

$$\text{(a)} \qquad\qquad \text{(b)} \qquad\qquad \text{(c)}$$

where $J_{i,k}$ denotes the 3D coordinates of the $i$-th joint in the $j$-th frame, with $i \in [1,k], j \in [1,n]$ $k = 15$. The number of matrix rows represents the duration of an action execution. Keeps the number of columns fixed, and increasing or decreasing the number of rows can slow

down or speed up the execution of an action. In our method, an average linear interpolation algorithm is applied to supplement the new frames of the original skeleton sequences, a randomly discarded approach is adopted to reduce the number of frames. As shown in Formula (1), matrix (b) is a new action sequence. It is provided by using the average linear interpolation algorithm to randomly insert a frame between the 1st frame and $n$-th frame of the original sequence a, and $Ni = (J_{i,m} + J_{i,m+1})/2, m \in (1, n)$. Moreover, matrix (c) is obtained by randomly discarding a frame.

### 3.2.2. Mimicking a Certain Action That Was Executed by People of Different Sizes

The advantage of skeleton-based HAR is that it can focus on the performance of human actions and ignoring human physical characteristics and background information. Furthermore, regardless of whether a human skeleton is estimated by RGB images or captured from depth cameras and the age or gender of a person, its structure is similar. Accordingly, we can scale up or down the human skeleton to produce samples of people of different sizes performing the same action. In other words, we can imitate a certain action executed by people of different sizes, which is a simple but effective method to generate skeleton samples. This method can be easily implemented by multiplying the original skeleton sequence by a suitable value. With reference to actual experience, we set the interval of this value to [0.5, 1.5].

### 3.2.3. Mimicking a Certain Action That Was Conducted in Different Places of the Field of Viewpoint

Similar to the second method, skeleton data are more robust in changing the field of viewpoint and occlusion situations. This means that the motion information of the skeleton joints is not significantly affected by the change of perspective. Thus, the new skeleton sequences can be provided by changing the position where an action was performed. Its physical significance mimics a certain action conducted in different places of the field of viewpoint. The method can be enacted by adding or subtracting a value from the entire skeleton sequence of an action.

### 3.3. Feature Extraction

According to the classical theory of physics, describing the motion of an object requires three elements, distance, speed, and direction. Accordingly, relative coordinate, immediate displacement and immediate motion orientation corresponding to three action elements are adopted to describe human motions. Compared with absolute motion data, these three kinds of relative features are viewpoint invariant and have better generalization abilities.

### 3.3.1. Relative Coordinate

For a skeleton sequence $S$ in the form of 3D coordinates with $N$ frames, where $S = \{F_1, F_2, \ldots, F_N\}$. Assume $J_{i,t}$ is the original coordinate of the $i$-th joint of the skeleton in frame $F_t$. $J_{spine,1}$ is the spine joint of the 1st frame and is adopted as a reference point. Each joint is defined as $J = (x, y, z)$. The relative coordinate $R = (x, y, z)$ can be calculated as follows:

$$R_{i,t} = J_{i,t} - J_{spine,1} \quad t \in [1, N], \ i \in [1, 15] \tag{2}$$

where $R_{i,t}$ represents the relative coordinates of the $i$-th joint in the $t$-th frame of the skeleton sequence $S$ and 15 is the number of joints. Compared with the original coordinates, the relative coordinates proposed in our method are viewpoint invariant. This means that the relative data are only related to the starting position of the action and will not be affected by the position of the camera. Moreover, the relative coordinate is determined by the difference between the coordinate of the $t$-th frame and the spine coordinate of the first frame. So, it not only depicts the spatial relations of human skeleton joints but also provides long-term temporal information.

### 3.3.2. Immediate Motion Orientation

Another geometric feature used in our method is immediate motion orientation. Suppose $\overrightarrow{Vi,t} = (x, y, z)$ is a 3D vector of joint $i$ in frame $t$. First, we map the 3-dimensional vector to the three planes $xy$, $xz$, and $yz$ to obtain three 2-dimensional vectors. Then, the immediate motion orientation can be obtained by calculating the cosine angles of the three 2-dimensional vectors on the corresponding coordinate axis. The movement orientation $O = (\cos \theta x, \cos \theta y, \cos \theta z)$ can be calculated as follows:

$$\begin{aligned} \cos \theta x &= \frac{x}{\sqrt{\Delta x^2 + \Delta y^2}} \\ \cos \theta y &= \frac{y}{\sqrt{\Delta x^2 + \Delta y^2}} \\ \cos \theta z &= \frac{z}{\sqrt{\Delta y^2 + \Delta z^2}} \end{aligned} \tag{3}$$

where $\theta$ is the angle between the motion vectors and the coordinate axis, and $\Delta x, \Delta y, \Delta z$ are the displacements of joint $i$ on the $x, y, z$ axis, respectively. In particular, the values of the immediate motion of the 1st frame are initialized to zero.

### 3.3.3. Immediate Displacement

It is not sufficient to describe human actions using only geometrical features. For example, it is difficult to distinguish "sitting down" and "standing up", whose spatial motion information is similar, so temporal information is important to HAR. Although relatively coordinated with long-term dynamic information, it lacks local motion information. Therefore, we employ immediate displacement to capture the local temporal features. In Formula (4), $D_{i,t+1}$ represents the immediate displacement of the $i$-th joint in the $t$-th frame, and $N$ is the length of the action sequence. Immediate displacement $D = (x, y, z)$ can be obtained by differencing between the original coordinates of the $t+1$-th and $t$-th frames. Similarly, the values of immediate displacement of the 1st frame are initialized to zero.

$$D_{i,t+1} = J_{i,t+1} - J_{i,t}, t \in [1, N-1], i \in [1, 15] \tag{4}$$

### 3.4. Image Encoding

In contrast to greyscale images, color image coding can imitate the apperceptive ability of the human visual system to extract more information and effectively promote the detailed texture patterns obscured in the images [19]. Consequently, the spatial-temporal motion information is encoded into color motion images in our proposed method. The concrete steps are as follows: First, the three relative features are extracted from each frame of an action sequence and these spatial-temporal features are combined into a feature matrix. In the following, a normalization function is selected to normalize the values of each feature matrix to chromatic RGB values. Although normalization functions, such as log () and tanh (), are widely employed to normalize data, they have an obvious flaw. They obscure the differences between large numbers. For instance, 1000 are 100 times 10, whereas log (1000) is slightly larger than log (10). Therefore, to better preserve the difference between large numbers, the maximum and minimum normalization function is used to take the place of conventional normalization. The data normalization is implemented using the following equations:

$$\overline{P} = \frac{P - \min(P)}{\max(P) - \min(P)} * (g_{max} - g_{min}) \ P \in (R, D, O) \tag{5}$$

where $P$ denotes the original feature matrix, $\overline{P}$ denotes the normalized feature matrix. On the left side of the multiplication sign, the values are normalized to 0–1. Then, these feature values are converted to RGB values by multiplying $(g_{max} - g_{min})$, where $[g_{min}, g_{max}]$ is the range of greyscale values. Generally, $g_{max} = 255$ , $g_{min} = 0$ is set as a full-scale transformation to a color image.

Finally, the three normalized feature matrices $\overline{R}, \overline{D},$ and $\overline{O}$ are converted to greyscale images separately. After which each greyscale image corresponds to an RGB channel to merge into a color image. Motion images with a size of only $45 * N$, where N is the frame number of an action and 45 means that the motion information of each joint is stored in 3 pixels. Although the joint-motion-data-based image is small in size and simple in structure, it not only contains spatial information but also carries short-term and long-term temporal features.

### 3.5. Network Model

According to [24,25], as the motion color image is small in size and simple in structure, a shallow CNN framework is enough to extract the deep features of the generated motion images for HAR. Moreover, according to the experimental results, the CNN models with complex structures did not perform ideally, because the size of the motion image proposed in our method is much smaller than the image size predesigned by the classical CNN model. For example, VGG19 specifies that the size of the input is $224 * 224$. After the convolution operation of multiple convolutional layers, the feature maps fed into the fully connected layers are quite small. Feature maps that are too small will result in overfitting and reduce the generalization ability of the network. Simply adjusting the image size to meet the input requirement of CNN models will damage the continuity of action; for example, walks may be incorrectly classified as runs. In addition, the authors [26] showed that CNN models could improve performance by balancing network depth, width, and image resolution. However, if magnification increases, the improvement decreases.

Based on the above analyses, a shallow CNN model is designed to extract the deep features of the generated motion images. Figure 3 shows the detail structures. Our proposed model consists of two convolutional layers with a kernel size of $3 * 3$, two max pooling layers with a kernel size of $2 * 2$ and three fully connected layers. In particular, rectified linear activation (ReLu) is employed in our network. Compared with saturation nonlinear functions, such as sigmoid, ReLu has a faster convergence rate, which can overcome the vanishing gradient problem and decrease the computing time [27]. In addition, two batch normalization layers have been added between the convolutional layer and max pooling layer, because it can normalize the output of each layer to have a mean value of 0 and variance of 1, which speeds up the training and convergence of the network. Furthermore, the batch normalization layer could replace the Dropout and L2 regularization [28]. For it has a regularization function to prevent overfitting and maintain the generalization ability of the network.

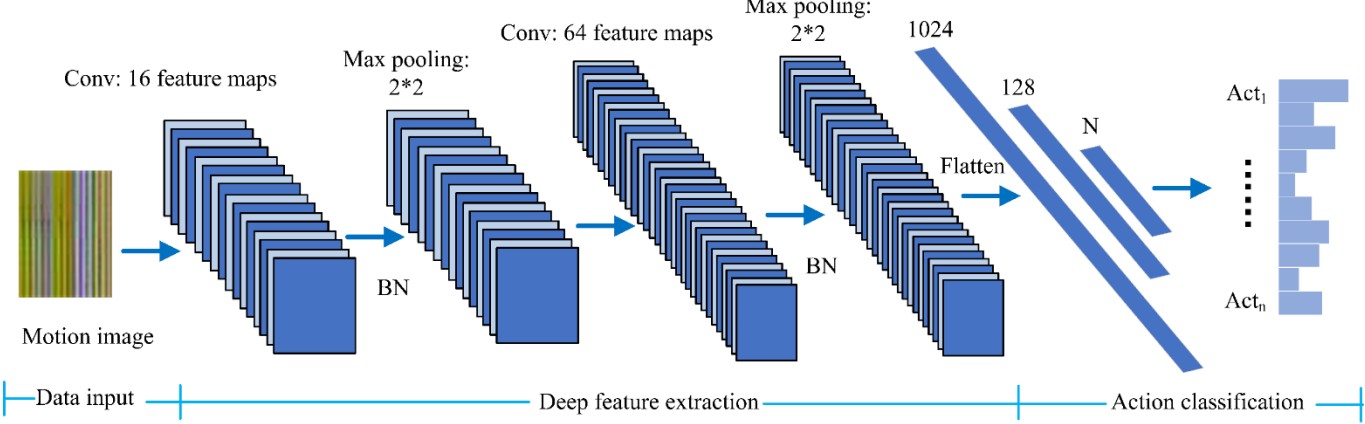

**Figure 3.** The CNN model is introduced for human action recognition in our proposed method. Conv represents convolutional layer, BN means the batch normalization layer, and N is the number of action categories.

Compared with classical CNN models, our model has two advantages. First, it can be easily trained from scratch without any pretraining. Second, it can achieve competitive experimental results with less time complexity and space complexity.

## 4. Experiment

In this section, the performance of our method is evaluated on two small-scale but challenging datasets, the Florence 3D Actions dataset [29] and UTkinect-Action 3D dataset [30] (after here use Florence-3D and UTkinect-3D instead). The first experiment is designed to explore the optimal frame number of skeleton sequences. Then, the effectiveness and general applicability of the data augmentation methods and the superiority of the relative coordinates proposed in this paper are verified. Finally, compared with the state-of-art methods, the experimental results show that our method achieves a competitive performance.

All models are trained with the same batch size = 128, training epochs = 100, optimizer (Adam with a fixed learning rate of 0.001), and cross entropy loss function of PyTorch [31]. On the two datasets, we follow the standard leave-one-out cross validation configuration, in which ten percent of samples are used as the test set and the remaining samples are the training set. The experiments are implemented on a desktop computer with an NVIDIA GTX-3060 GPU and Intel Core i7-11700KF 3.60 GHz processor.

### 4.1. Datasets

Florence-3D: The dataset includes a total of 215 activity samples obtained by a Kinetic camera. It consists of nine activities such as wave, drink from a bottle, answer phone, clap, tight lace, sit down, stand up, read, watch, and bow. One of the above actions is repeated 2–3 times by 10 subjects, and each object is labelled with 15 joints.

UTkinect-3D: A single stationary Kinect is used to collect this dataset, and there are 10 action categories such as walk, sit down, stand up, pick up, carry, throw, pull, push, wave hands, and clap hands. Each action was performed twice by 10 subjects, and each subject had 20 joints, 15 of which were selected in our method. Because one sample of the skeleton sequence information was lost, only 199 samples were available in this dataset.

### 4.2. Comparison of Different Frames of Skeleton Sequence

As mentioned in Section 3.2.1, an action sequence refinement strategy is adopted to unify the length of the skeleton sequence. In this way, we can obtain fixed size images for CNN model training. The resolution of the images generated by the action sequences with different frames is also different, which could also affect the accuracy of the results. Consequently, this experiment is designed to select the optimal number of frames, N, for the skeleton sequence. In addition, the average number of frames of the Florence-3D is approximately 20, and for UTkinect-3D there are 30 frames. Consequently, the frame numbers are initialized to 30.

As shown in the Figure 4, there are recognition results of multiple classical CNN models and our model training on images generated by data augmentation with different frame numbers on two datasets. In the charts, the X-axis represents the frame number, and the Y-axis is the classification accuracy. When the number of frames, N, is 30, the classification result of each model is at the lowest point, such that the accuracy of LeNet is 76.74% and 81.24% on the two datasets, respectively. As the number of frames increases, the accuracy of the models also increases until N = 60, reaches the highest value, and then the accuracy drops somewhat. Hence, the frame number N of each skeleton sequence is unified to 60 and the size of the generated motion image is fixed to 45 ∗ 60.

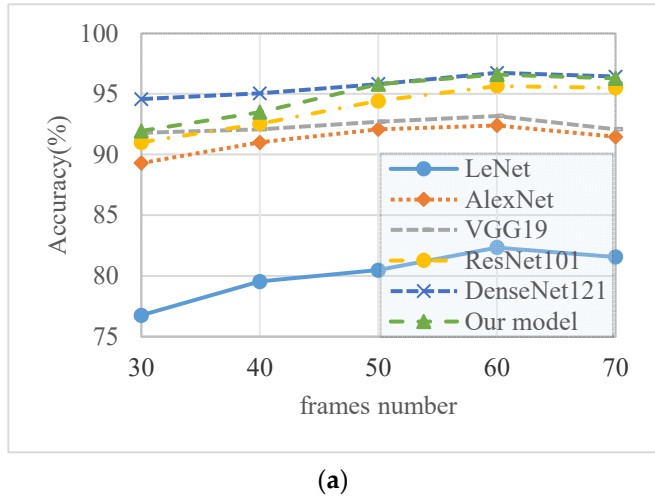

(**a**)

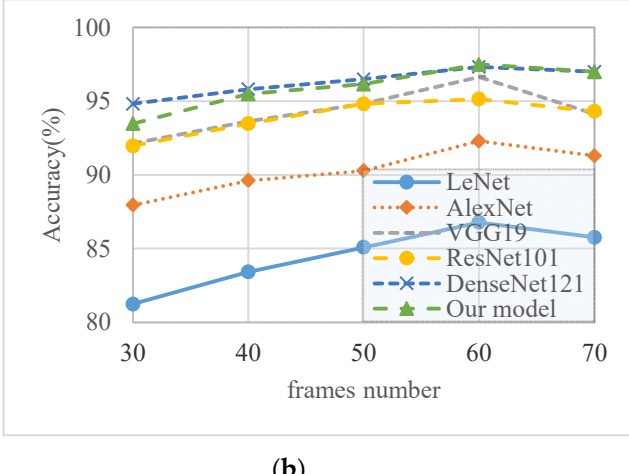

(**b**)

**Figure 4.** The recognition accuracy of multiply CNN model training on images generated by data augmentation with different frame numbers on the Florence-3D dataset (**a**) and UTkinect-3D dataset (**b**).

In detail, when the number of frames, N = 60, then DenseNet121 and our model achieves excellent and similar results on two datasets. On Florence-3D, our model achieves an accuracy of 96.59%, 0.15% lower than DenseNet121. Our model is 0.16% higher than DenseNet121 on UTkinect-3D. Although our model achieves the second-best accuracy, our model is still adopted as the final model. Because, as shown in Table 1, the space complexity of our model is 10.25 MB less than DenseNet121, the time complexity is only approximately one over ten of DenseNet121. This means our model consumes less memory and has higher efficiency than DenseNet121 [32,33].

**Table 1.** Comparison of space complexity and time complexity of CNN models.

| Networks | Space Complexity | Time Complexity |
|----------|------------------|-----------------|
| LeNet | 1.14 MB | 2.19 MFLOPs |
| AlexNet | 213.42 MB | 4.02 GFLOPs |
| VGG19 | 333.88 MB | 11.99 GFLOPs |
| ResNet101 | 259.83 MB | 1.86 GFLOPs |
| DenseNet121 | 41.94 MB | 150.39 MFLOPs |
| Our Model | 31.69 MB | 13.97 MFLOPs |

### 4.3. Demonstrate Effectiveness of Data Augmentation

Figure 5 shows the training curves of different CNN models based on the original samples of two datasets. In addition, Table 2 shows the test accuracy of different CNN models trained on the original images and the generated images of two datasets. Specifically, for the original sample, we use the method of inserting repeated frames and randomly discarding frames to unify the frame number of skeleton sequences to 60. Although inserting repeated frames will cause information redundancy, it could retain as much original motion information as possible. Combining the two figures, it can be seen that CNN models have achieved acceptable results in the training stage. However, overfitting appears in the test stage, and as the number of network layers increases, it becomes more serious. Especially for DenseNet121, it reaches an accuracy rate of over 90% in the training stage on two datasets. However, in the testing stage, its accuracy is only 40.91% and 50%, respectively.

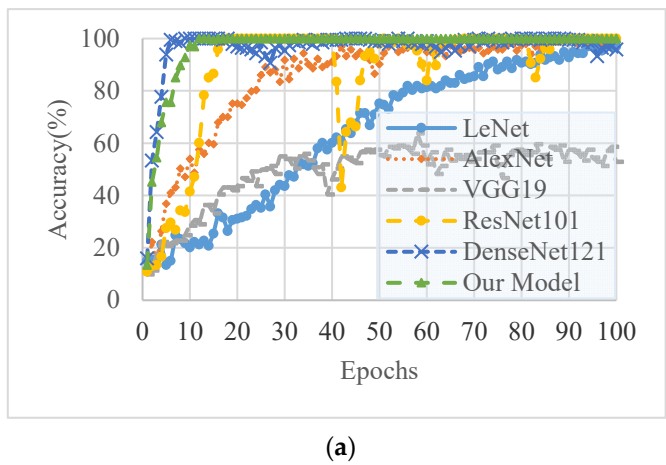
(**a**)

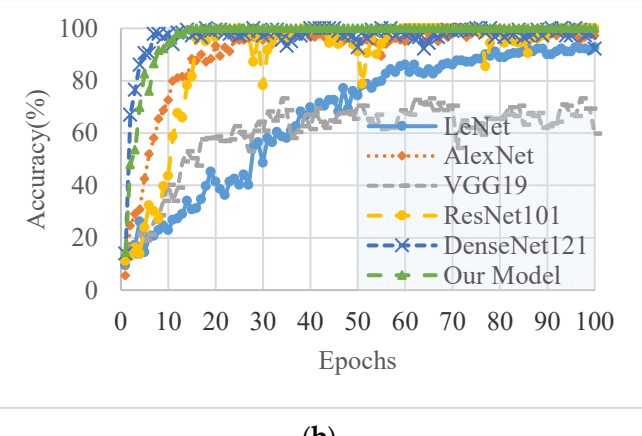
(**b**)

**Figure 5.** Training process of multiple CNN models based on original samples of Florence-3D (**a**) and UTkinect-3D (**b**).

**Table 2.** Comparison of the test accuracy of different CNN models on original images and generated images of Florence-3D dataset and UTkinect-3D dataset.

| Dataset | Score Item | LeNet | AlexNet | VGG19 | ResNet101 | DenseNet121 | Our Model |
|---|---|---|---|---|---|---|---|
| Florence-3D | Original samples | 77.27% | 69.7% | 54.56% | 47.73% | 40.91% | 72.73% |
| | Generated samples | 82.33% | 92.4% | 93.18% | 95.66% | 96.74% | 96.59% |
| | Improved accuracy | 5.06% | 22.7% | 38.62% | 47.93% | 55.83% | 23.86% |
| UTkinect-3D | Original samples | 75% | 72.5% | 65% | 52.5% | 50% | 75% |
| | Generated samples | 86.77% | 94.56% | 96.64% | 95.14% | 97.32% | 97.48% |
| | Improved accuracy | 11.77% | 22.06% | 31.64% | 42.64% | 47.32% | 22.48% |

As shown in Table 2, the recognition accuracy of the CNN model training on generated images has significantly improved. Specifically, the test accuracy of our model has improved by 23.86% and 22.48% on two datasets, respectively, and the test results of the other models also have improved to varying degrees. For example, the test results of LeNet separately increased from 77.27% to 82.33% on Florence-3D and improved from 75% to 86.77% onUTkinect-3D. Additionally, DenseNet121 has the biggest improvement, with a boost of 55.83% and 47.32% on two datasets separately. The experimental results show that the overfitting problem caused by a lack of sufficient training samples has been alleviated effectively by training on the enlarged dataset. Besides, the data augmentation methods proposed in this paper are effective and can be widely adopted in skeleton-based action recognition methods.

### 4.4. Comparison of Original Coordinates and Relative Coordinates

To validate the superiority of the relative coordinates proposed in this paper, the original coordinate is used to form a new feature combination with an immediate displacement and immediate motion orientation. After that the new feature combination are encoded into motion images similarly. It can be seen that Table 3 shows that the first feature combination achieves test accuracies of 93.64% and 93.13% on the two datasets separately. Compared with the former, the feature combination adopted in our method is improved by 2.95% and 4.35%, respectively. It shows that the relative coordinate proposed in this paper is more general applicability, because the relative coordinate is only related to the starting position of action but is not affected by the position of the camera. In addition, compared with the original coordinate, the relative coordinate could provide more accurate spatial relations of human skeleton joints and long-term dynamical information.

**Table 3.** Compare the recognition results of original coordinate and relative coordinate combine with other two features separately on two datasets using our network (coord represents coordinate of joints and disp, orien denote the immediate displacement and immediate motion orientation).

| Dataset | Feature Combination | Accuracy |
|---------|---------------------|----------|
| Florence-3D | original coord + disp + orien | 93.64% |
| | relative coord + disp + orien | 96.59% |
| UTkinect-3D | original coord + disp + orien | 93.13% |
| | relative coord + disp + orien | 97.48% |

### 4.5. Comparison of the Normalization Function

As mentioned in Section 3.4, the log () function, as a popular normalization function, is adopted in numerous methods. Nevertheless, it does not fit our proposed method. In our method, $\overline{P} = \frac{\log(P)}{\log(max(P))} * (g_{max} - g_{min})$ $P \in (R, D, O)$ was adopted to normalize the data. Similar to Formula (5), we normalize the values and then convert them into pixel values. As seen from Table 4, the recognition results of the generated images normalized by log () were inferior to the generated images normalized by maxmin (), representing the function in (5). We believe the reason for this problem can be explained by the following. The log () function is typically adopted for scenarios with large data distributions, mapping data to a concentrated interval. However, it is not suitable for our method because the amplitude of the human actions in a scene is not particularly large and the log () function will obscure the differences in action between frames, leading to a sharp decline in identification accuracy. Rather, maxmin () could overcome the drawback by retaining the motion information sufficiently in the normalization process. Therefore, the maxmin () function is used as the normalization function in our proposed method.

**Table 4.** Compare the recognition results of two normalization function on two datasets using our network.

| Normalization Function | Florence-3D | UTkinect-3D |
|------------------------|-------------|-------------|
| Log () | 85.74% | 89.94% |
| Maxmin () | 96.59% | 97.48% |

### 4.6. Comparison of State-of-the-Art Methods

Florence-3D: The proposed method achieves a recognition accuracy of 96.59%, taking the lead for the performance competition given in Table 5. In [34], the authors directly employed original joint coordinates as spatial features and then used Bag of Words approach to encode features. In [35], Cai et al. extracted geometric features from 3D skeleton data to model skeleton human action information in the group space, and a CNN model was used for learning and classification. In [36], the authors considered each body joint as a kinematics sensor to calculate the Linear Joint Position Feature (LJPF) and Angular Joint Position Feature (AJPF) to encode complex human actions. Both of the above approaches propose rich geometric features to characterize actions, but they ignore the importance of the temporal dimension information.

**Table 5.** Compared with the state-of-the-art methods on Florence-3D dataset.

| Methods | Accuracy |
|---------|----------|
| Lie group + CNN [35] | 93% |
| Skeletal BoW [34] | 94.34% |
| Learing feature combonation [37] | 94.39% |
| HDS-SP [38] | 95.88% |
| Kinematics posture feature [36] | 96% |
| Our method | 96.59% |

UTkinect-3D: The overall recognition accuracy results on UTkinect-3D are presented in Table 6. It can be seen that our proposed method achieves the second-best performance. In [39], Zhang et al. input original joints and several other geometric features into a multistream LSTM for recognition, after fusing the score to obtain the final result. In [40], a skeleton-based graph structure was applied to represent human motion data. Nevertheless, the topology of the graph is fixed over all layers, only local features can be obtained and it lacks collaboration information for long distances. In [41], the authors proposed a new action recognition scheme to describe human motion information more accurately and robustly. The scheme concatenated relative joint position (RJP), joint velocity and joint acceleration to form a spatial-temporal representation. Dynamic time warping (DTW) and Fourier temporal pyramid (FTP) are employed to handle the rate variations and noise in the skeleton data, respectively.

**Table 6.** Compared with the state-of the art methods on UTkinect-3D dataset.

| Methods | Accuracy |
| --- | --- |
| Featurs fusion + MLSTM [39] | 96% |
| GFT [40] | 96% |
| CNN with Attention Mechanism [42] | 96.1% |
| Kinectic features + 1 DCNN [22] | 96.9% |
| ST-representation [41] | 97.59% |
| Our method | 97.48% |

*4.7. Discussion*

Although our method achieves competitive results, it still has shortcomings. Figures 6 and 7 describe the recognition results of each action on Florence-3D and UTkinect-3D using our method separately. As shown in Figure 6, answer phone and read watch are the two actions with the lowest recognition, at only 88.06% and 86.89%, respectively. We consider the reason for this phenomenon to be that without considering external objects, some actions are so similar that they will be misclassified. For example, answer phone may be classified as drink from bottom because both movement processes are raising one hand to the head. The key poses are exhibited in Figure 8. For the action, the read watch swings one hand to the chest, similar to the swing of two hands on the chest when clapping, so the test result is not ideal. Similarly, in Figure 7, the action throw is incorrectly detected as pull. From the perspective of skeleton sequence, both actions push one hand far away, but judging from the images, things fly out of the hand during the throw actions.

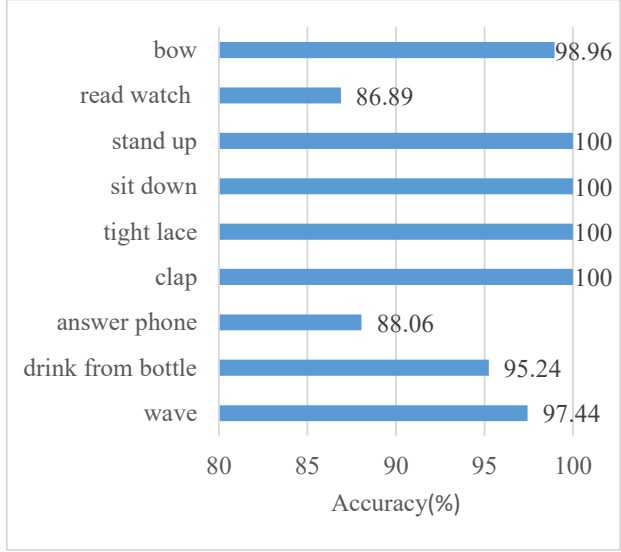

**Figure 6.** The recognition accuracy of each action on Florence-3D dataset using our method.

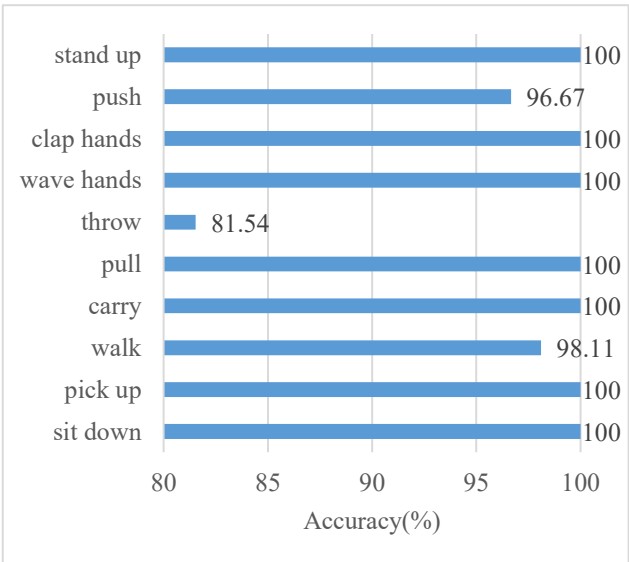

**Figure 7.** The recognition accuracy of each action on UTkinect-3D dataset using our method.

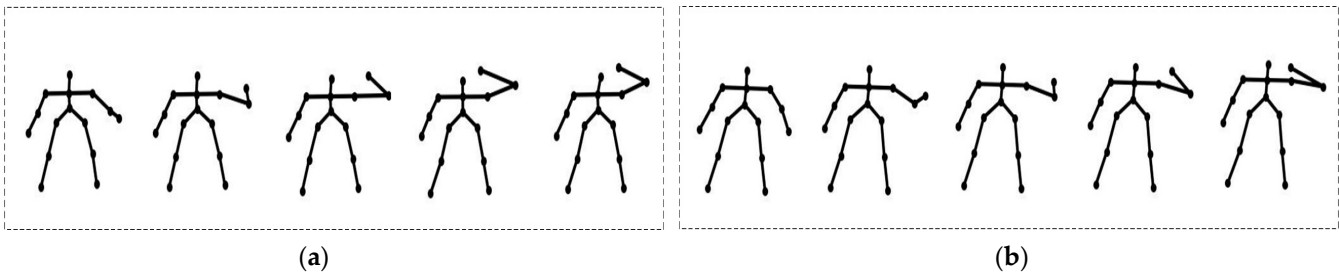

| (**a**) | (**b**) |

**Figure 8.** Five key poses of two similar actions in the Florence-3D. (**a**) Drink from bottom. (**b**) Answer phone.

In summary, the recognition of some actions is not good enough due to a lack of consideration of the influence of external objects on human motions. Consequently, to further improve the recognition accuracy of our method, we detect external object information that is related to the action and encode it into an image as a feature.

## 5. Conclusions

In this paper, we proposed a novel human action recognition approach in which skeleton sequences are transformed into color motion images, and a shallow CNN model is introduced for action classification. For each action sequence, three relative features with informative spatial-temporal information are calculated to depict human motions. Next, these features are normalized and transformed into color images. The motion image contains the spatial relations of human skeleton joints, and abundant dynamic information is provided to represent an action. Finally, a shallow CNN model is adopted to extract the deep features of the generated motion images and classify actions. The experimental results show that our model has superior efficiency and a high capacity. The research contributions of our method can be summarized in three points. First, a novel relative coordinate not only depicts the spatial relations of human skeleton joints but also provides long-term temporal information, is introduced in this paper. Moreover, the relative feature is viewpoint invariant and has more general applicability. Second, an effective skeleton sequence refinement strategy is adopted to cope with varying-length action sequences. Third, a data augmentation strategy is proposed to provide action samples and prevent CNN models from overfitting when training on a few samples. Experimental results prove that our data augmentation methods can effectively solve the overfitting problem and can be widely adopted in skeleton-based HAR methods.

In our future work, we will focus on more efficient relative features and encoding techniques for transforming action sequences into color motion images. In addition, we would like to detect external object information related to human poses and encode them into images as a feature for performance improvements.

**Author Contributions:** J.C.: conceptualization, methodology, software, writing—original draft. W.Y.: supervision, writing—reviewing and editing. C.L.: supervision, reviewing, and editing. L.Y.: supervision, reviewing, and editing. All authors have read and agreed to the published version of the manuscript.

**Funding:** This research was supported by the Special Project 03 of Jiangxi Provincial Department of Science and Technology under Grant 20212ABC03A36, by the Scientific and Technological Projects of the Nanchang Science and Technology Bureau under Grant GJJ202010, GJJ202017, GJJ191004 and by the National Natural Science Foundation of China under Grant 61862043.

**Institutional Review Board Statement:** Not applicable.

**Informed Consent Statement:** Not applicable.

**Data Availability Statement:** Not applicable.

**Conflicts of Interest:** The authors declare no conflict of interest.

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
