# Peer review of "A Data Augmentation Method for Skeleton-Based Action Recognition with Relative Features"

_applsci, doi:10.3390/app112311481_

Round 1
Reviewer 1 Report
The authors propose a method for skeleton-based human action recognition. Relative coordinates are used to obtain spatial and long-term temporal information. A data augmentation method enlarges the small size of training samples. Additionally, generated color images train a shallow CNN model to prevent overfitting. Two skeleton datasets test the proposed method. The method achieves good results and performs like state-of-the-art methods.
The presentation of your method is understandable, because of Fig. 1. In general, the text sometimes is barely understandable. You should consider a major revision of the writing style for your paper.
E.g.:
Lines 174ff: “Accordingly, we can scale up or down human skeleton to produce samples of people with different sizes performing the same action, and it can imitate a certain action is executed by peoples with different sizes, which is a simple but effective method to generate skeleton samples.”
Lines 319ff: “In detail, the test accuracy of our model has improved by 23.86% and 22.48% on two datasets respectively, while the test results of other models also have improved to varying degrees, for example, the test results of LeNet separately elevated from 77.27% and 75% to 82.33% and 86.77% and DenseNet121 has the biggest improvement, boosting by 55.83% and 47.32% on two datasets separately.”
Some tips for your revision:
- Consider scientific writing style
- Use shorter sentences
- Check grammar: some sentences do not have a verb or an understandable structure
- Reduce excessive use of “while” throughout the paper
- Consider defining abbreviations and variables in a dedicated section
Reviewer 2 Report
In this paper, human action recognition algorithm is proposed for depth camera images.
1. The authors stated that one of main contribution is the use of relative coordinate. However, I believe the use of relative coordinate is rather standard. For example, see [A].
2. The log(.) function is a popular normalization function. It would be desirable if authors show that the function in (5) is better than log(.) function by comparing two results.
[A] H. Rahmani and M. Bennamoun, "Learning Action Recognition Model from Depth and Skeleton Videos," 2017 IEEE International Conference on Computer Vision (ICCV), 2017, pp. 5833-5842, doi: 10.1109/ICCV.2017.621.
Round 2
Reviewer 1 Report
The present version is not a major revision as recommended in the first review. So far, three text passages have been adjusted. These text passages have been improved according to an acceptable, scientific writing style. This improvement should be done with the whole publication. The passages given in the last review have been given as examples of overly complicated and incomprehensible sentences. Therefore, a major revision is still suggested, as the publication should not be published in its present form.
Reviewer 2 Report
I think the paper has been improved according to reviewers' suggestions.
Author Response
Thank you very much for your efforts and advices.
Round 3
Reviewer 1 Report
The content presentation has improved significantly. The content is now much more understandable. Good job! On page 18, three lines of text are hidden behind the table. Some figures separate coherent paragraphs. With these minor formatting adjustments, the paper can be published as is. To my experience, the formatting is additionally controlled by the MDPI-editors and adjusted before publication.